# Effects of Storage Temperature, Packaging Material and Wash Treatment on Quality and Shelf Life of Tartary Buckwheat Microgreens

**DOI:** 10.3390/foods11223630

**Published:** 2022-11-14

**Authors:** Huiling Yan, Wenfei Li, Hongxu Chen, Qingxia Liao, Mengying Xia, Dingtao Wu, Changying Liu, Jianxiong Chen, Liang Zou, Lianxin Peng, Gang Zhao, Jianglin Zhao

**Affiliations:** 1Key Laboratory of Coarse Cereal Processing, Ministry of Agriculture and Rural Affairs, Sichuan Province Engineering Technology Research Center of Coarse Cereal Industrialization, School of Food and Biological Engineering, Chengdu University, Chengdu 610106, China; 2Huantai Biotechnology Company Ltd., Chengdu 610213, China

**Keywords:** Tartary buckwheat microgreens, storage temperature, packaging material, wash treatment, postharvest quality

## Abstract

Tartary buckwheat microgreens (TBM) are popular worldwide products but display an extremely short shelf life. Thus, the effects of storage temperature, packaging material, and wash treatment on the quality and shelf life were analyzed. Headspace composition, weight loss, electrolyte leakage, microbial population and sensory quality were investigated during storage. Results showed that shelf life and quality of TBM decreased with the increment of storage temperature when stored at 5–25 °C. During 5 °C storage, LDPE bags were the best packaging materials for preserving the quality of LDPE, PE and HDPE bags. On the basis of 5 °C and LDPE packages, ClO_2_ + citric acid wash treatment could further inhibit quality deterioration and extend the shelf life. The results demonstrated bioactive constituents and antioxidant capacity were significantly affected by storage time. The study provides insights into developing optimal packaging and storage conditions for TBM.

## 1. Introduction

Microgreens, also known as “tender immature greens” or “salad crop shoots”, are cultivated from the seeds of vegetables, herbs or crops and harvested after 10 to 20 days of seed germination [1,2,3]. As the living standards have risen, tens of microgreens have appeared in upscale markets, specialty grocery stores, and restaurants and gained great popularity due to their excellent sensory and nutritional attributes over the past decades [2,4]. However, microgreens are perishable and have an extremely short shelf life during ambient storage for 1–2 days [5]. Moreover, there is an increasing food safety concern since microgreens are mainly consumed raw [6].

Temperature is one of the most significant storage factors for the quality and shelf life of postharvest microgreens [7]. Studies in various microgreens, such as radish microgreens, buckwheat microgreens, and mustard microgreens, demonstrated the importance of temperature control in postponing postharvest quality deterioration and extending shelf life via reducing weight loss, atmosphere component and growth of food spoilage-associated microorganisms [7,8,9]. Moreover, optimal storage temperature varies for different kinds of microgreens, for example, 1 °C for radish microgreens and 5 °C for buckwheat (*Fagopyrum esculentum*) microgreens [7,9], indicating the cruciality of storage temperature selection.

The atmosphere component is another important storage factor affecting the characteristics of postharvest microgreens [7,9,10]. The package atmosphere of microgreens is determined by several elements, among which packaging film selection is a convenient and effective way to keep freshness, extend shelf life and protect microgreens from food spoilage-associated microorganisms and other environmental pollutants [7,9,10]. However, information about packaging film selection for microgreens is still scarce.

In addition, water washing is an important processing step protecting fresh-cut products from contamination, which helps to remove dust, debris and cell exudates caused by the harvesting procedure and kill microorganisms. Conventionally, the washing process is performed with chlorinated water or organic acids, and its application has been reported in postharvest microgreens, such as buckwheat and radish microgreens [7,9,11]. Thus, systemic analysis of wash treatment on postharvest TBM also needs more attention.

Tartary buckwheat (*Fagopyrum tataricum*) is abundant in protein, lipids, vitamins, flavonoids and polyphenolic compounds, which are beneficial for human health due to the free radicals scavenging and oxidative damage reduction activities [12]. Tartary buckwheat sprouts and microgreens, germinated from Tartary buckwheat seeds, contain higher amounts of flavonoids and are gluten-free, which jointly make them widely spread in the world as healthcare products [13,14]. However, the supply of fresh Tartary buckwheat sprouts and microgreens in the food chain is still very limited due to their perishable characteristics and lack of postharvest techniques.

## 2. Materials and Methods

### 2.1. Sample Preparation and Packaging

Tartary buckwheat seeds (Cultivar Xiqiao No. 1 of *Fagopyrum tataricum*), supplied by the Coarse Cereal Processing Center of Chengdu University (Chengdu, Sichuan, China), were immersed for 12 h in deionized water (dH_2_O) at room temperature after rinsing in dH_2_O till the liquid is transparent. Afterward, the seeds were sown in the soil in a growth chamber at 22 ± 1 °C. The seeds were kept in darkness for the first 4 days and then illuminated by light-emitting diodes (LEDs) with the intensity of 50 μmol m^−2^ s^−1^ (16/8 h, light/dark). After 12 days of cultivation, the samples were harvested, and microgreens without blemishes and mechanic injuries were used for further analysis.

### 2.2. Temperature, Packaging and Wash Treatments

For all the treatments, microgreens were kept in bags of 20 cm × 30 cm with 30 g per bag. For temperature analysis, microgreens were packed in polyethylene (PE) (12 μm) bags. Samples were stored at 5, 10, 15, 20, or 25 °C for 8 days and were periodically analyzed on 0, 2, 4, 6 and 8 d. For packaging material analysis, microgreens were sealed in PE (12 μm), low-density polyethylene (LDPE) (38 μm) and high-density polyethylene (HDPE) (6.8 μm) bags, respectively. Samples were stored at 5 °C for 8 days and evaluated on 0, 2, 4, 6 and 8 d. For wash treatment analysis, microgreens were divided into four groups randomly. The first three groups were immersed in tap water, chlorinated water (2 mg L^−1^ ClO_2_) and citric acid solution (1% citric acid (*w*/*v*)) for 10 min, respectively. The fourth group was firstly immersed in chlorinated water (2 mg L^−1^ ClO_2_) for 5 min and then in citric acid solution (1% citric acid (*w*/*v*)) for 5 min. After being air-dried, microgreens were sealed in LDPE bags and stored at 5 °C for 12 days and evaluated on 0, 4, 8 and 12 d.

### 2.3. Analysis of Packaging Headspace Atmosphere Composition

A gas analyzer (CYES-II, NANBEI Instrument Co. Ltd., Zhengzhou, China) was used to measure the contents of CO_2_ and O_2_ in the headspace of TBM in the bags according to the instruction of the manufacturer. 2 mL of headspace air was injected into the analyzer slowly. The content of O_2_ was recorded when the gas flow was stopped, and the content of CO_2_ was recorded till the value of the analyzer was stable.

### 2.4. Overall Quality and Off-Odor Analysis

The overall quality and off-odor analysis were performed by a six-member trained panel according to the method of Xiao et al. [15]. The overall visual quality was evaluated with a 9-point hedonic scale, where 9, 5 and 1 represent “like extremely”, “neither like nor dislike”, and “dislike extremely”, respectively [16]. Off-odor was determined by a 0–4 scale, where 0, 1, 2, 3 and 4 denote the extent of off-odor representing “no”, “slight”, “moderate”, “strong”, and “extremely strong”, respectively.

### 2.5. Weight Loss and Electrolyte Leakage Analysis

Weight loss was performed by scaling the microgreens before and after storage. Weight loss was computed according to the following formula:Weight loss (%) = (m_0_ − m_1_)/m_0_ × 100%

m_0_ = the weight of microgreens before storage, g; and m_1_ = the weight of microgreens after storage, g.

Electrolyte leakage was analyzed according to the method of Xiao et al. [7] with minor modifications. Firstly, TBM (3 g) was soaked in distilled water (150 mL) for 30 min at 25 °C. Then, samples were frozen at −20 °C for 24 h and then thawed at 25 °C with tap water. The electrical conductivity of the solution was detected with a conductivity meter (model DDS-307A; Shanghai INESA & Scientific Instrument Co. Ltd., Shanghai, China) before freezing and after thawing. Relative electrolyte leakage was calculated by the following formula:Relative electrolyte leakage (%) = G_0_/G_1_ × 100%

G_0_ = the electrical conductivity of the sample before freezing, and G_1_ = the electrical conductivity of the sample after thawing.

### 2.6. Microbial Counts

Microbial enumeration of microgreens was carried out according to the method of Chun and Song [17] with minor modifications. Briefly, 25 g of TBM were put into 225 mL of peptone water (0.1% sterile peptone, *w*/*v*), homogenized and diluted serially by peptone water. 1 mL sample of appropriate dilution was spread on Plate Count Agar (PCA, Difco Co., Berkaa, Lebanon) and Potato Dextrose Agar (PDA, Difco Co., Berkaa, Lebanon) to determine the counts of total coliforms and yeasts and molds, respectively. PCA plates were incubated at 37 °C for 48 h, while PDA plates were incubated at 28 °C for 72 h. Each microbial count was performed in triplicates and expressed as log CFU (colony forming units) g^−1^.

### 2.7. Determination of Total Phenolics, Total Flavonoids and Main Phenolic Compounds

Tartary buckwheat microgreens were dried in a drying oven at 50 °C to a constant weight, powdered, filtered by a sieve with 60 mesh and stored at −20 °C for future analysis. A weight of 0.2 g powder of dried microgreens was put into 12 mL of methanol solution (60%, *v*/*v*) and subjected to ultrasound (200 W) for 30 min at 50 °C. Subsequently, samples were centrifuged at 8000 rpm for 10 min. The supernatant was collected for future analysis.

Total phenolic content was detected by the colorimetric method according to the method of Singleton and Rossi [18] with Folin-Ciocalteu’s phenol reagent. The content was expressed as g gallic acid equivalents per kg dry weight (g GAE kg^−1^).

Total flavonoid contents were detected by aluminum nitrate colorimetric assay according to the formation of a complex flavonoid-aluminum, exhibiting a maximum absorbance at 510 nm [19]. Rutin was used as a standard compound. 1 mL of sample or rutin standard solution was added into a 10 mL volumetric flask, mixed with 2.5 mL of distilled water and 0.15 mL of 5% NaNO_2_ for 6 min; after that, 0.3 mL of 10% Al(NO_3_)_3_ was added and reacted for another 6 min. A volume of 2 mL of 4% NaOH was added to stop the reaction, and the mixture was kept in the dark for 15 min. The absorbance was detected at 510 nm. The content was expressed as g rutin equivalents per kg dry weight (g RE kg^−1^).

The main phenolic compounds were measured according to the method of Lee et al. [20] with minor modifications. The supernatant was filtered through a disposable syringe filter (PTFE, 0.45 μm, hydrophobic; Advantec, Tokyo, Japan), and the filtrate was measured by an HPLC system (LC-20A; Shimadzu Co., Kyoto, Japan) equipped with an InertSustain-C18 column (4.6 mm × 150 mm, 5 μm). The analytical conditions were as follows: column oven temperature, 45 °C; UV detector: 350 nm; injection volume: 10 μL; flow rate: 1.0 mL min^−1^; solvent system, a mixture of (A) MeOH: water: acetic acid (5:92.5:2.5, *v*/*v*/*v*) and (B) MeOH: water: acetic acid (95:2.5:2.5, *v*/*v*/*v*); gradient program, 20% to 36% solvent B, from 0 to 23 min, 36% to 60% solvent B, from 23 to 26 min, 60% solvent B, from 26 to 34 min, 20% solvent B, 34.1 min, 20% solvent B, from 34.1 to 40 min. The content of each phenolic compound was quantified based on HPLC peak areas and calculated as equivalents of seven respective standard compounds. All phenolic contents were expressed as g per kg dry weight (g kg^−1^).

### 2.8. Determination of Antioxidant Capacity

The antioxidant capacities of TBM were performed with diphenylpicrylhydrazyl (DPPH, %) and 2,2′-azino-bis (3-ethylbenzothiazoline-6-sulfonic acid) (ABTS, %) [21,22]. About 0.2 g of TBM powder was extracted in 80% ethanol and centrifuged at 12,000 rpm for 30 min.

For ABTS, an equal volume of 7 mM ABTS and 2.45 mM potassium persulphate were mixed and incubated in darkness at room temperature for 12–16 h. After incubation, the ABTS stock was diluted till the absorbance was 0.7 ± 0.002 at 734 nm. Sample: 0.2 mL of diluted extract was mixed with 3 mL of ABTS solution; Control: 0.2 mL of Trolox was mixed with 3 mL of ABTS solution. The mixtures were incubated in the dark at room temperature for 10 min and detected at 734 nm. The ABTS scavenging activity was expressed as mmol Trolox per kg dry weight (mmol kg^−1^ DW).

For DPPH, sample: 0.5 mL of diluted extract was added into 4 mL of DPPH solution; control: 0.5 mL of diluted extract was added into 4 mL of DPPH solution; blank: 0.5 mL of methanol was added into 4 mL of DPPH solution. The mixtures were incubated in the dark at room temperature for 30 min and detected at 517 nm.
DPPH scavenging activity (%) = (1 **−** (A_1_
**−** A_2_)/A_0_) × 100%

A_0_: absorbance of blank; A_1_: absorbance of sample; A_2_: absorbance of control.

### 2.9. Statistical Analysis

All the experiments were performed in three replicates by a totally randomized design. Data were analyzed via a one-way analysis of ANOVA at a significance level of 0.05 using SPSS version 26.0 (SPSS Inc., Chicago, IL, USA) at a significance level of 0.05. Data were presented as the mean ± SE.

## 3. Results and Discussion

### 3.1. Effect of Storage Temperatures on Quality and Shelf Life of TBM

The headspace atmosphere composition of packed TBM was notably (*p* < 0.05) influenced by storage temperature and time (Figure 1A,B). All the packed microgreens exhibited a rapid decrease in O_2_ concentration in the first 2 d storage and then kept a relatively stable O_2_ concentration till the end of the storage (Figure 1A). Conversely, the CO_2_ concentration of all the packed microgreens increased dramatically in the initial 2 d storage and then maintained at a nearly stable level till the end of storage (Figure 1B). Moreover, O_2_ in packed microgreens stored at 5 °C was less depleted than those stored at 10 °C, 15 °C, 20 °C and 25 °C during the whole storage, which might be due to the inhibitory effect of low temperature on respiratory rate [23]. These results were in agreement with the reports on various microgreens or sprouts, such as buckwheat microgreens, radish microgreens and mung bean sprouts [7,9,23].

All the packed microgreens lost weight during the storage, and the microgreens stored at higher temperatures displayed a higher rate of weight loss (Figure 1C). As for storage at 5 °C and 10 °C, weight loss rose up to 1.6% and 1.9% at 8 d, while those of storage at 15 °C and 20 °C increased to about 2.1% and 4.3% at 4 d already (Figure 1C). These are in accordance with findings of previous studies [23], indicating low-temperature storage could inhibit respiration rate, resulting in slower weight loss [24].

Electrolyte leakage is an indicator of cell membrane integrity, which could be affected by ripening, stress or mechanical injury [25]. In the study, electrolyte leakage of microgreens was significantly influenced by the storage temperature (Figure 1D). There was an obvious drop (significant at *p* < 0.05) in electrolyte leakage of microgreens stored at all temperatures during the initial storage, which was also reported in radish microgreens and buckwheat microgreens [7,9]. Moreover, electrolyte leakage of microgreens stored at 15 °C, 20 °C and 25 °C decreased more than those of microgreens stored at 5 °C and 10 °C (Figure 1D), indicating that storage temperature might affect the recovery process of a mechanical injury caused by harvest. In addition, microgreens stored at 5 °C and 10 °C showed higher electrolyte leakage than those stored at 15 °C (Figure 1D), suggesting that TBM stored at 5 °C and 10 °C might be susceptible to chilling injury during prolonged storage.

The growth of total coliforms and yeast and mold (Y&M) was significantly affected by storage temperature and time (Figure 1E,F). The count of total coliforms in TBM increased along with the storage in all treatments (Figure 1E). Total coliforms counts of microgreens stored at 25 °C increased more rapidly than those of microgreens stored at 5 °C, 10 °C, 15 °C and 20 °C (Figure 1E). Noteworthily, total coliforms counts of microgreens stored at 25 °C reached around 7.3 log CFU g^−1^ after 2 d storage, while those of microgreens stored at 5 °C and 10 °C exhibited 6.9 and 7.2 log CFU g^−1^ after 8 d storage (Figure 1E). The population of Y&M in microgreens exhibited a similar trend as those of total coliforms. These results indicated that low temperature inhibited the growth of coliforms and Y&M (Figure 1F), which is consistent with the findings of other microgreens or sprouts [7,9,23].

The overall quality and off-odor scores are vital parameters for determining the marketability and popularity of fresh products. Similarly, overall quality and off-odor changes of TBM were notably influenced by storage temperature and time (Figure 1G,H). During the whole storage, the overall quality of microgreens stored at 5 °C ranked highest on each sampling day, followed by those stored at 10 °C, 15 °C, 20 °C and 25 °C serially. There was a sharp decrease in the overall quality of microgreens stored at 25 °C (around 5.5) during the initial 2 d storage, while the overall quality of microgreens stored at 5 °C decreased much more slowly and displayed an overall quality of about 5.7 after 8 d of storage. Off-odor scores of microgreens stored at all temperatures increased during storage, and the higher the storage temperature was, the stronger the off-odor was measured (Figure 1H). After 2 d, TBM stored at 20 °C, and 25 °C exhibited moderate off-odor (score 1.3 and 1.8, respectively), while only slight off-odor was detected in microgreens stored at 5 °C, 10 °C and 15 °C (score 0.5, 0.5 and 0.8, respectively) (Figure 1H). An off-odor score of microgreens stored at 20 °C increased to 2.3 after 4 d storage, while that of microgreens stored at 5 °C and 10 °C came to 2.4 and 2.5 after 8 d storage, respectively (Figure 1H). The development of off-odor was highly related to the O_2_ concentration reduction due to undesirable fermentation and microbial growth, which agreed with the findings on buckwheat microgreens and baby spinach [9,26].

These findings demonstrated that storage temperature is crucial for quality preservation and safety control of the delicate TBM. Microgreens stored at 5 °C exhibited the longest shelf life and optimum quality. Thus, 5 °C was chosen as the ideal storage temperature for TBM and used for the subsequent experiments.

### 3.2. Effect of Packaging Materials on the Quality of TBM Stored at 5 °C

The headspace atmosphere composition of packed TBM stored at 5 °C was notably (*p* < 0.05) influenced by packaging materials (Figure 2A,B). Packages packed with LDPE films exhibited a significantly more rapid reduction of O_2_ and an increase of CO_2_ than those packed with PE and HDPE films. The O_2_ levels in packages packed with LDPE films decreased dramatically during the initial 2 d storage and then decreased relatively slowly during the later storage, while those of packages packed with HDPE and PE films decreased rapidly during the first 2 d storage and maintained at a relatively stable state till the end of storage except an obvious reduction occurred from 6 d to 8 d in packages packed with PE films (Figure 2A). In addition, the CO_2_ levels in all the packages reached an equilibrium around 1.5–2.0 kPa after the rapid increase in the initial 2 d storage and then maintained the state till the end of storage (Figure 2B).

All the packed microgreens lost weight during the storage, and microgreens packed with HDPE bags exhibited the highest rate of weight loss, followed by those of microgreens packed with PE bags, while those of microgreens packed with LDPE bags were the least (Figure 2C).

A significant difference was observed in the electrolyte leakage of TBM packed by different packaging films (Figure 2D). The electrolyte leakage of microgreens packed by LDPE bags was lower than those of HDPE and PE-packed microgreens. A sharp decrease occurred in the electrolyte leakage of microgreens packed by all the packaging films from 0 d to 6 d, followed by an obvious but relatively small increase in microgreens packed by LDPE bags from 6 d to 8 d, while the electrolyte leakage of those packaged by PE and HDPE bags just fluctuated slightly around 4.0 ± 0.1% during this period.

The initial microorganisms counts of packed microgreens (both total coliforms and Y&M) were less than those reported in radish microgreens and mung bean sprouts [7,23]. Even though the packaging materials obviously affected the atmosphere compositions in the packed microgreens, no significant difference was observed among counts of total coliforms on microgreens packed by all the packaging materials and only a slight difference was detected among counts of Y&M on 4 d and 8 d (Figure 2E,F). These results indicated that the growth of total coliforms and Y&M of TBM were not affected by packaging materials when stored at 5 °C, which was similar to the findings on radish microgreens and fresh-cut cilantro leaves [7,27]. These data suggest that temperature is the paramount factor determining the growth of the great majority of microorganisms.

The overall quality and off-odor changes of TBM were notably affected by packaging materials and storage time when stored at 5 °C (Figure 2G,H). Generally, scores of visual quality decreased gradually during the storage in all treatments, while those of off-odor increased along the storage. TBM packed by LDPE bags maintained the best visual quality during the entire storage duration, followed by those in PE and HDPE bags sequentially. The off-odor score in microgreens packed by PE bags was the lowest during the initial 2 d storage, but a sharp increase occurred from 2 d to 4 d, resulting in the strongest off-odor detected in the later storage compared to those packed by LDPE and HDPE bags.

These results indicated that there was no markable difference in keeping the quality and extending the shelf life of TBM among all the packaging materials, which was in accordance with the report on radish microgreens [7]. However, relatively better quality was observed after 8 d storage at 5 °C in microgreens packed by LDPE bags, with the lowest electrolyte leakage and weight loss and the best visual quality score. In this case, LDPE bags were selected for further analysis of wash treatment on TBM.

### 3.3. Effect of Wash Treatment on the Quality of TBM Packed in LDPE Bags Stored at 5 °C

The headspace atmosphere composition of packaged TBM was significantly affected by wash treatments (Figure 3A,B). There was more O_2_ and less CO_2_ in packages of water -washed microgreens during the storage compared to those washed with ClO_2_, citric acid and ClO_2_ + citric acid. The difference might be due to the higher respiration rate of microgreens induced by wash procedures with ClO_2_, citric acid and ClO_2_ + citric acid.

From 0 d to 8 d storage, there was no significant difference in weight loss of TBM observed among all the wash treatments (Figure 3C). From 8 d to 12 d storage, the weight loss rate of water -washed microgreens was the highest, indicating that washing with chemical disinfectants or organic acids was beneficial to reduce the weight loss rate during the later storage of packaged TBM.

The electrolyte leakage of microgreens was significantly influenced by wash treatments and storage time (Figure 3D). The initial electrolyte leakage of microgreens washed by citric acid and ClO_2_ + citric acid was slightly higher than those of microgreens washed by water and ClO_2_. From 0 d to 8 d storage, the electrolyte leakage of microgreens in all the wash treatments fluctuated in a range of 1.5 to 3.5, while that of ClO_2_ washed microgreens increased sharply to 8.7 from 8 d to 12 d, followed by 7.0, 5.3 and 4.5 of microgreens washed by water, citric acid and ClO_2_ + citric acid, respectively. These results indicated that citric acid and ClO_2_ + citric acid might play a positive role in slowing the quality deterioration of TBM during later storage.

The overall growth of coliforms and Y&M on microgreens were notably affected by wash treatments and storage time (Figure 3E,F). The growth of total coliforms and Y&M on microgreens were relatively slow during the initial 8 d storage and increased obviously from 8 d to 12 d storage. During the entire storage, microgreens washed by water and ClO_2_ exhibited more coliforms and Y&M than those washed by citric acid and ClO_2_ + citric acid, which was in accordance with the effect of those treatments on electrolyte leakage. Jointly, these results indicated that citric acid and ClO_2_ + citric acid might maintain the tissue integrity of microgreens by inhibiting the growth of coliforms and Y&M.

There were no obvious changes in visual quality and off-odor in all the treated microgreens during the initial 4 d storage (Figure 3G,H). The visual quality decreased slightly from 4 d to 8 d storage to 8 and decreased rapidly to about 4.5 from 8 d to 12 d storage, while the off-odor score increased to around 3 with a relatively stable rate from 4 d to 12 d storage. Considering the obvious decrease of visual quality and development of off-odor in the initial 4 d storage, and lower visual quality (around 5.5) and higher off-odor (around 2.7) scores after 8 d storage in unwashed microgreens (Figure 2G,H), the results indicated that these wash treatments played a positive role in maintaining the postharvest quality of TBM. Consistent with the results of electrolyte leakage, the microgreens washed by citric acid and ClO_2_ + citric acid maintained better visual quality and lower off-odor development, revealing that visual quality loss and off-odor development of postharvest TBM was highly related to tissue deterioration as reported in radish microgreens [7].

These results indicated that wash treatment could play a positive role in preserving the quality and elongating the shelf life of TBM. Especially microgreens washed with ClO_2_ + citric acid exhibited the best sensory quality, lowest electrolyte leakage and microorganisms during the whole storage. Thus, the changes in bioactive compounds and antioxidant activity during postharvest storage in TBM washed with water and ClO_2_ + citric acid were analyzed in the following study.

### 3.4. Changes in Bioactive Compounds and Antioxidant Activity of TBM during Postharvest Storage

During the 12 d storage, the total phenolics content of TBM in the two treatment groups changed in similar trends, and there was no significant difference found between the two groups (Figure 4A,B). Phenolic compounds are important secondary metabolites, and the accumulation of phenolic compounds in the plant is significantly affected by environmental stresses [28]. ClO_2_ + citric acid -washed microgreens exhibited higher contents of total phenolics and flavonoids after wash treatment, implying ClO_2_ + citric acid induced the accumulation of phenolic compounds in microgreens (Figure 4A,B). Total phenolics in microgreens increased during the first 8 d storage and then decreased during the later storage, indicating that low temperature and mechanic injury caused by harvest and wash treatment induced the accumulation of total phenolics. The content of total flavonoids in microgreens of both groups was maintained at a relatively stable level during the initial 4 d storage and decreased with an increasing rate during the later storage. ClO_2_ + citric acid -washed microgreens exhibited higher content of total flavonoids than those washed by water during the whole storage.

As reported by Lee et al. [20], chlorogenic acid, orientin, isoorientin, vitexin, isovitexin, rutin and quercetin are the main phenolic compounds in Tartary buckwheat sprouts. Thus, the content of these compounds in TBM was detected via LC-MS analysis during storage (Table 1). Similar to the changes of total flavonoids in microgreens during storage, the content of these compounds was maintained in the initial 4 d storage and then decreased till the end of storage in both groups. Moreover, ClO_2_ + citric acid -washed microgreens also exhibited higher contents of all these compounds during the whole storage.

ABTS and DPPH radical scavenging capacity were notably affected by storage time but not wash treatment (Figure 4C,D). The trends of ABTS and DPPH radical scavenging capacity were both similar to that of total phenolics, indicating that phenolic compounds were the main antioxidants in TBM during storage. However, the changes in antioxidant activities were not consistent with those of individual flavonoid compounds. Interestingly, both results were also found in radish microgreens during storage [15], indicating that further studies in changes of individual bioactive constituents are needed to uncover the mechanism of antioxidant activities in microgreens during storage.

## 4. Conclusions

In the study, various postharvest processing circumstances on the quality and shelf life of TBM were investigated. Results indicated the quality of harvested TBM was significantly affected by storage temperature, followed by wash treatment and packaging material sequentially. Storage temperature significantly affected the changes in O_2_ and CO_2_ composition, microbial growth, visual quality and off-odor development during storage. The packaging materials significantly influenced the changes in O_2_ and CO_2_ composition, weight loss rate and electrolyte leakage. However, there were no major differences in overall quality among package materials until the end of storage. Wash treatment notably affected the sensory quality and microbial growth, especially ClO_2_ + citric acid washing. As a flavonoid-rich food, contents of total phenolics, total flavonoids and individual flavonoid compounds, and antioxidant capacity increased or fluctuated slightly during the initial 8 d storage and declined sharply from 8 d to 12 d storage. Change patterns of antioxidant capacity were similar to those of total phenolics but not individual flavonoid compounds. These results demonstrated that combined 5 °C storage, LDPE bags and ClO_2_ + citric acid wash treatment is a promising processing procedure to prolong the shelf life and preserve the quality of TBM. Further studies are needed to illustrate the regulation mechanism of the procedure on the antioxidant capacity of TBM.

## Figures and Tables

**Figure 1 foods-11-03630-f001:**
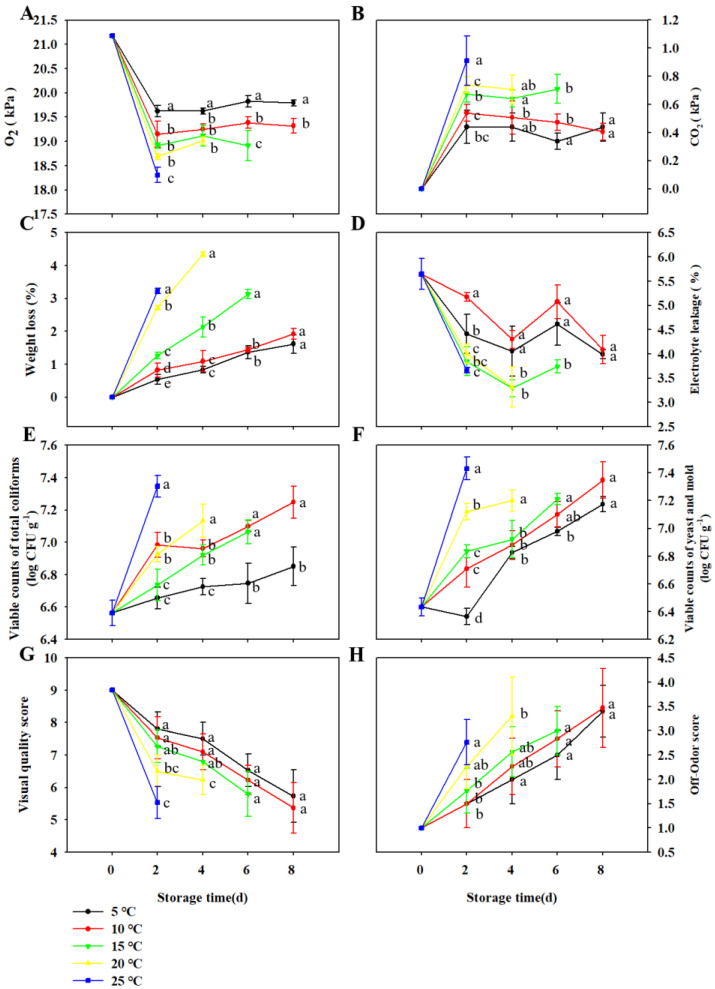
Effect of storage temperature on the changes in O_2_ (**A**) and CO_2_ (**B**) partial pressures within packages, weight loss (**C**), electrolyte leakage (**D**), counts of total coliforms (**E**) and yeast and mold (Y&M) (**F**), overall quality score (**G**) and off-odor score (**H**) of TBM packed by PE bags. The vertical bar represents ± standard error. Significant differences between samples (within the same time point) are indicated with different lowercase letters above the plots.

**Figure 2 foods-11-03630-f002:**
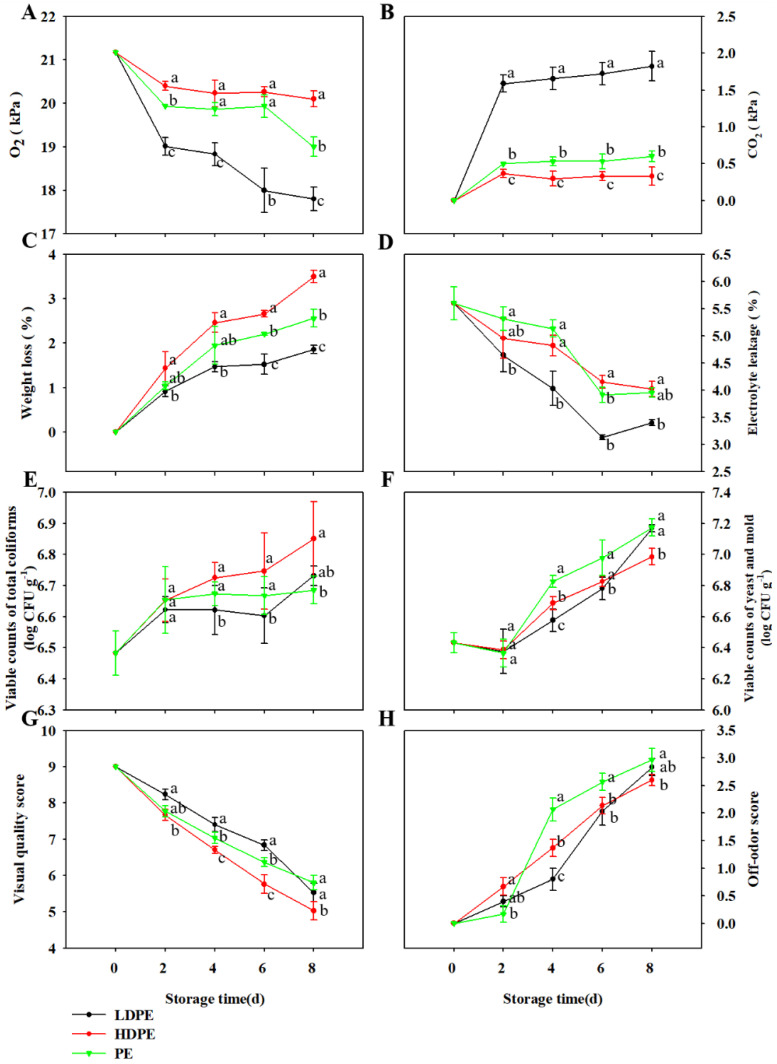
Effect of packaging material on the changes in O_2_ (**A**) and CO_2_ (**B**) partial pressures within packages, weight loss (**C**), electrolyte leakage (**D**), counts of total coliforms (**E**) and yeast and mold (Y&M) (**F**), overall quality score (**G**) and off-odor score (**H**) of TBM during 5 °C storage. The vertical bar represents ± standard error. Significant differences between samples (within the same time point) are indicated with different lowercase letters above the plots.

**Figure 3 foods-11-03630-f003:**
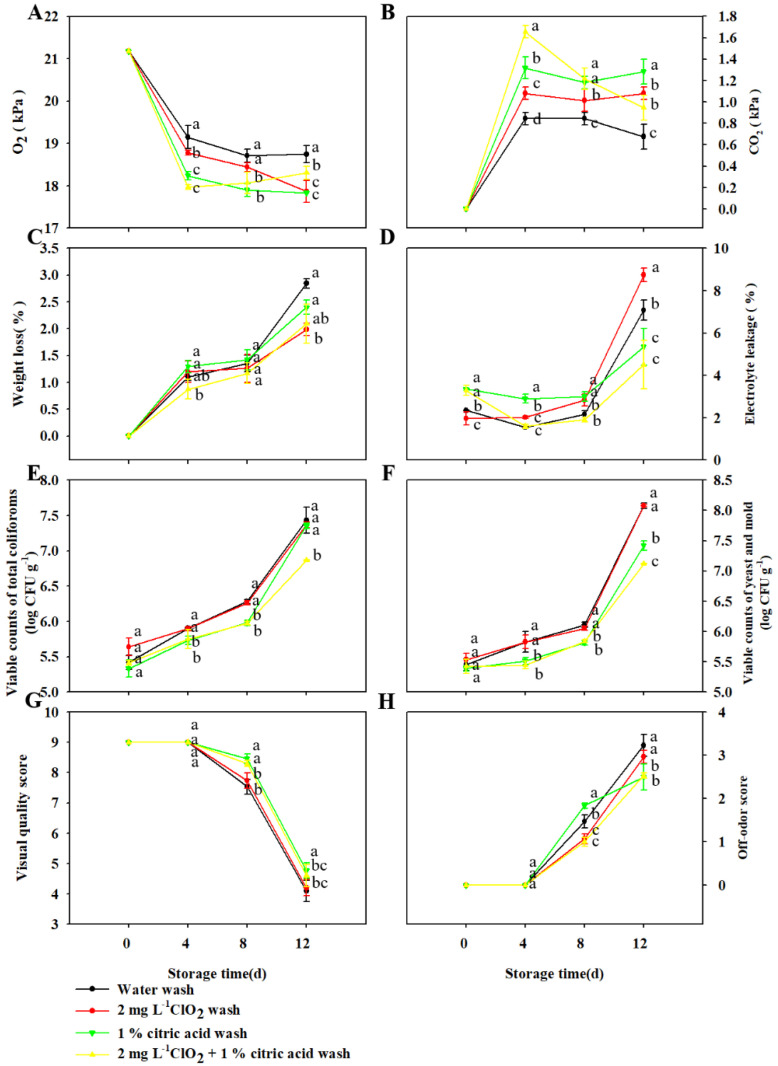
Effect of wash treatment on the changes in O_2_ (**A**) and CO_2_ (**B**) partial pressures within packages, weight loss (**C**), electrolyte leakage (**D**), counts of total coliforms (**E**) and yeast and mold (Y&M) (**F**), overall quality score (**G**) and off-odor score (**H**) of TBM packed by LDPE bags during 5 °C storage. The vertical bar represents ± standard error. Significant differences between samples (within the same time point) are indicated with different lowercase letters above the plots.

**Figure 4 foods-11-03630-f004:**
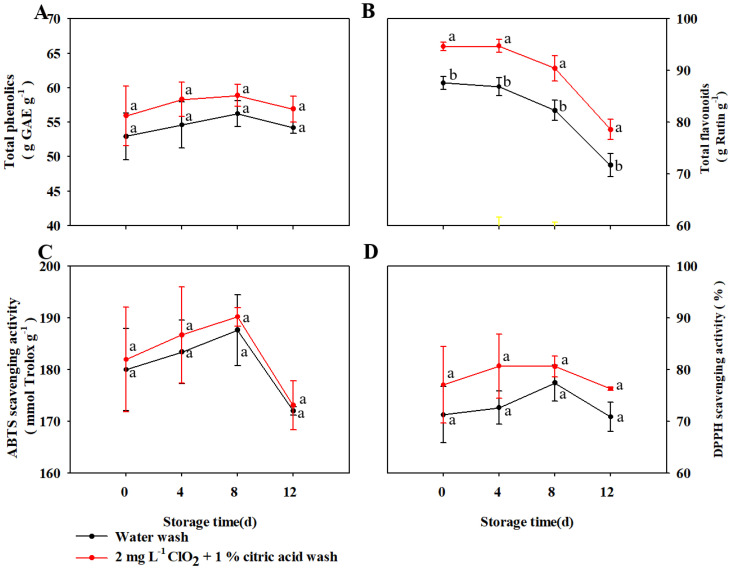
The changes in contents of total phenolics (**A**) and total flavonoids (**B**), and 2,2′-azino-bis (3-ethylbenzothiazoline-6-sulfonic acid) (ABTS) (**C**) and Diphenylpicrylhydrazyl (DPPH) (**D**) radical scavenging activity of TBM packed by LDPE bags during 5 °C storage. The vertical bar represents ± standard error. Significant differences between samples (within the same time point) are indicated with different lowercase letters above the plots.

**Table 1 foods-11-03630-t001:** Changes in the contents of major phenolic compounds.

Storage Time (d)	Chlorogenic Acid (g kg^−1^)	Orientin (g kg^−1^)	Isoorientin (g kg^−1^)	Vitexin (g kg^−1^)	Isovitexin (g kg^−1^)	Rutin (g kg^−1^)	Quercetin (g kg^−1^)
Water/0	1.29 ^b^ ± 0.05	0.18 ^a^ ± 0.02	0.42 ^b^ ± 0.01	1.19 ^a^ ± 0.02	1.70 ^b^ ± 0.04	78.53 ^a^ ± 0.85	0.61 ^a^ ± 0.16
ClO_2_ + ctric acid/0	1.38 ^a^ ± 0.09	0.20 ^a^ ± 0.03	0.48 ^a^ ± 0.02	1.24 ^a^ ± 0.08	1.78 ^a^ ± 0.10	79.34 ^a^ ± 2.09	0.63 ^a^ ± 0.21
Water/4	1.26 ^b^ ± 0.01	0.17 ^a^ ± 0.01	0.44 ^b^ ± 0.01	1.18 ^b^ ± 0.01	1.68 ^b^ ± 0.02	78.36 ^a^ ± 0.65	0.61 ^a^ ± 0.09
ClO_2_ + ctric acid/4	1.61 ^a^ ± 0.00	0.19 ^a^ ± 0.00	0.49 ^a^ ± 0.00	1.25 ^a^ ± 0.01	1.79 ^a^ ± 0.01	79.61 ^a^ ± 0.40	0.60 ^a^ ± 0.11
Water/8	1.19 ^b^ ± 0.02	0.22 ^a^ ± 0.01	0.43 ^b^ ± 0.01	1.20 ^b^ ± 0.01	1.69 ^b^ ± 0.02	74.89 ^a^ ± 1.07	0.41 ^a^ ± 0.13
ClO_2_ + ctric acid/8	1.50 ^a^ ± 0.01	0.23 ^a^ ± 0.00	0.49 ^a^ ± 0.00	1.29 ^a^ ± 0.01	1.83 ^a^ ± 0.02	76.53 ^a^ ± 0.79	0.40 ^a^ ± 0.09
Water/12	1.06 ^b^ ± 0.02	0.17 ^b^ ± 0.03	0.41 ^b^ ± 0.00	1.16 ^a^ ± 0.00	1.61 ^b^ ± 0.01	64.15 ^b^ ± 0.64	0.52 ^a^ ± 0.12
ClO_2_ + ctric acid/12	1.34 ^a^ ± 0.01	0.21 ^a^ ± 0.02	0.47 ^a^ ± 0.01	1.22 ^a^ ± 0.02	1.76 ^a^ ± 0.03	66.76 ^a^ ± 1.32	0.55 ^a^ ± 0.07

Different lowercase letters indicated significant differences between samples (within the same time point).

## Data Availability

Data are contained within the article.

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
