# Peer review of "Effects of Storage Temperature, Packaging Material and Wash Treatment on Quality and Shelf Life of Tartary Buckwheat Microgreens"

_foods, 2022, doi:10.3390/foods11223630_

Round 1

Author Response

Dear Editor and Reviewers, 

     Please find our responses in the attached word file.

Best regards,

Jianglin Zhao

Reviewer 2 Report

Manuscript “Effects of Storage Temperature, Packaging Material and Wash Treatment on Quality and Shelf Life of Tartary Buckwheat Microgreens” fits the journal's profile. It is spelled correctly; the purpose of the work is clearly formulated, the research methods are well described. The results are shown in the figures and in the table. The description of the axes in the figures is not legible, the font is too small. The conclusions are justified in the content of the work. However: why was the research conducted at high temperatures (20, 25 oC)? While already in the publication [9] it has been shown that 5 oC is the optimal temperature. This should be explained and justified in the manuscript introduction.

L180: ..accumulated dramatically… - it means?

L203- 205: why this is so?

Author Response

(The authors gave the same response as above.)
